# A molecular atlas of plastid and mitochondrial proteins reveals organellar remodeling during plant evolutionary transitions from algae to angiosperms

**Parth K. Raval** [ID]**, Alexander I. MacLeod, Sven B. Gould** [ID] *

Institute for Molecular Evolution, Heinrich-Heine-University Düsseldorf, Düsseldorf, Germany

* gould@hhu.de

**Data Availability Statement:** Supplementary figures are available in the supplementary information file. Additional supplementary data, in-house scripts and source data for the main and

## Abstract

Algae and plants carry 2 organelles of endosymbiotic origin that have been co-evolving in their host cells for more than a billion years. The biology of plastids and mitochondria can differ significantly across major lineages and organelle changes likely accompanied the adaptation to new ecological niches such as the terrestrial habitat. Based on organelle proteome data and the genomes of 168 phototrophic (Archaeplastida) versus a broad range of 518 non-phototrophic eukaryotes, we screened for changes in plastid and mitochondrial biology across 1 billion years of evolution. Taking into account 331,571 protein families (or orthogroups), we identify 31,625 protein families that are unique to primary plastid-bearing eukaryotes. The 1,906 and 825 protein families are predicted to operate in plastids and mitochondria, respectively. Tracing the evolutionary history of these protein families through evolutionary time uncovers the significant remodeling the organelles experienced from algae to land plants. The analyses of gained orthogroups identifies molecular changes of organelle biology that connect to the diversification of major lineages and facilitated major transitions from chlorophytes en route to the global greening and origin of angiosperms.

## Introduction

Fewer natural phenomena have been as transformative to planet Earth as the global greening through plants [1,2]. The proliferation of plants on land rests on the emergence and expansion of the Chloroplastida, also referred to as the Viridiplantae or simply the green lineage. The Chloroplastida are made up of 3 phyla: chlorophytes, streptophytes, and the prasinodermophytes that are thought to be the sister lineage to the 2 former [3]. Chloro- and prasinodermophytes are represented by algae only, whereas streptophytes are made up of algae and embryophytes, the latter uniting all land plants [3–5]. The list of key adaptations that fostered land plant expansion in a macroevolutionary context are multiple: roots, a mutualistic symbiosis with fungi, stomata, a cuticle, polyplastidy, and an expansion of many metabolite families such as flavonoids to name a few [1,3–10]. These innovations, evolving gradually in the

supplementary figures are available on Zenedo: https://zenodo.org/records/10855592.

**Funding:** We thank the Deutsche Forschungsgemeinschaft for grants awarded to SBG (SFB 1208-2672 05415 and SPP2237–440043394) and the Moore and Simons Initiative grant (9743) that was awarded to William Martin, who was so generous to financially support AIM. The funders had no role in study design, data collection and analysis, decision to publish, or preparation of the manuscript. https://www.dfg.de/ https://www.simonsfoundation.org/.

**Competing interests:** The authors have declared that no competing interests exist.

**Abbreviations:** ASR, ancestral state reconstruction; GOG, green orthogroup; HMM, hidden Markov model; KEGG, Kyoto Encyclopedia of Genes and Genomes; KOID, KEGG orthology identification; MOG, mitochondrial orthogroup; PAP, PEP associated protein; PEP, plastid-encoded RNA polymerase; PGI, phosphoglucose isomerase; POG, plastid orthogroup; RING, really interesting new gene; UV, ultraviolet.

common ancestor of land plants (LCA), provided a decisive fitness advantage over the non-terrestrial chloro-, prasinodermato-, and streptophyte algal relatives [1,11].

The eponymous organelle of plants, the chloroplast, underwent various changes, too. It adapted in multiple ways to the challenges characterizing the habitat the LCA encountered. Improving stress response was necessary to deal for instance with increased levels of ultraviolet (UV) high light stress and to cope with temperature shifts that change rapidly on land in contrast to in water [12–14]. Polyplastidy, a phenomenon that separates plastid from nuclear division, leading to cells that can harbor more than one plastid per cell, was part of being able to develop larger body plans [12,15,16]. To communicate stress and the need for component biosynthesis, an elaborate retrograde signaling evolved on the basis of messenger proteins such as GUN1 and maybe WHIRLY [17,18]. In combination, these adaptations were decisive for the success of streptophytes, which is evident in the number of species they have evolved and the sheer biomass they produce [1,19].

Plastids do not operate autonomously, but are part of an intricate metabolic network and even physically interact with other compartments such as the endoplasmic reticulum and peroxisomes [20,21]. Marked metabolic and physical interactions of plastids also concern the only other compartment of ancient endosymbiotic origin: the mitochondrion. Plant mitochondria are much less in the focus of plant research. Next to their canonical functions, they are known to be involved in immunity, lipid metabolism, and other (eco) physiological processes that are frequently in crosstalk with the photosynthetic organelle [22,23]. Like plastids, mitochondria were critical in the evolution and continued adaptation of important physiological traits, which characterize the green lineage. A notable example of preadaptation includes malate decarboxylation in the C4 photosynthetic pathway [24]—a trait of the green lineage [25] that improves plant photosynthetic efficiency in warm and dry habitats [26]. Similarly, some components of mitochondrial retrograde signaling also evolved in the land plants and likely contributed to its ROS and draught tolerance [27].

In spite of the importance of these 2 organelles of endosymbiotic origin in coordinating their duties, the evolution of components specific to chloroplast and mitochondrial biology has not been explicitly studied in light of streptophyte evolution or plant terrestrialization. Previous work has determined genes specific to certain plant clades and that are catalogued by valuable resources such as the GreenCut [28]. Such analyses, however, did not focus on organelle biology nor clustered protein families. They were also limited by a low number of archaeplastidal genomes and insufficient methods for orthology inference available at that time. Since then, genome assemblies of members from previously unsampled clades has increased manyfold [11,29–37] and more organelle proteomes and better functional annotations are available. Similarly, and concomitantly, the development of novel and accurate algorithms for orthology inference [38–41], along with advances in experimental biology allow to now identify critical evolutionary changes in an eco-evo context of plastid and mitochondrial biology that underpin the success of the Chloroplastida.

Here, we curate a database of protein families unique to the green lineage. We plot their evolution across the major splits in the evolutionary history of streptophytes, focusing on the biology of the 2 organelles of endosymbiotic origin. We report that the number of plastid- and mitochondria-associated protein families changes most significantly at 2 evolutionary bifurcations: firstly, at the green lineage itself and secondly at the split between Zygnematophyceae and embryophytes at the water to land transition. The newly recruited protein families influenced organellar processes such as carbon and lipid metabolism, information processing, and organelle development. We provide an extensive catalogue of the changes the proteomes of plastid and mitochondria experienced throughout streptophyte evolution, which offers

multiple angles from which to explore major evolutionary transitions such as the conquest of land and embryophyte diversification.

## Results

### Half of the chloroplastida protein families are unique to embryophytes

Out of a total of 12,862,035 proteins, 95% were categorized from 686 eukaryotes and grouped into 331,570 orthogroups (deposited on Zenodo [42]). From these, 31,650 were present only in chloroplastida and classified as green orthogroups (GOGs) (S1 Fig; [42]). An examination of GOG distribution among green species revealed that around half of all GOGs were unique to terrestrial plants (Fig 1A). Approximately 400 GOGs appeared in more than 90% of species, referred from here on to as the "core GOGs" (Fig 1B). For only 5% of all GOGs, a functional annotation could be identified (Fig 1C; [42]). For embryophyte-specific GOGs, the numbers were comparable, yet they maintained a consistent distribution of identified functions, including a substantial fraction of membrane trafficking and ubiquitination-related proteins (Fig 1D; [42]). Notably, for the core GOGs the number is higher. For 30% functional annotations covering photosynthesis, mitochondrial formation, trafficking, and information processing could be identified (Fig 1E; [42]). The functions for a vast majority of the GOGs remain elusive [42], numbers that mirror those of previous studies [28], and they hence provide an excellent ground for experimental exploration.

### Mitochondrial and plastid proteomes of the Chloroplastida expanded with the origin and diversification of the green lineage

To investigate changes in the proteomes of plastids and mitochondria, we curated 1,906 plastid and 825 mitochondrial orthogroups (POGs and MOGs, respectively) based on published

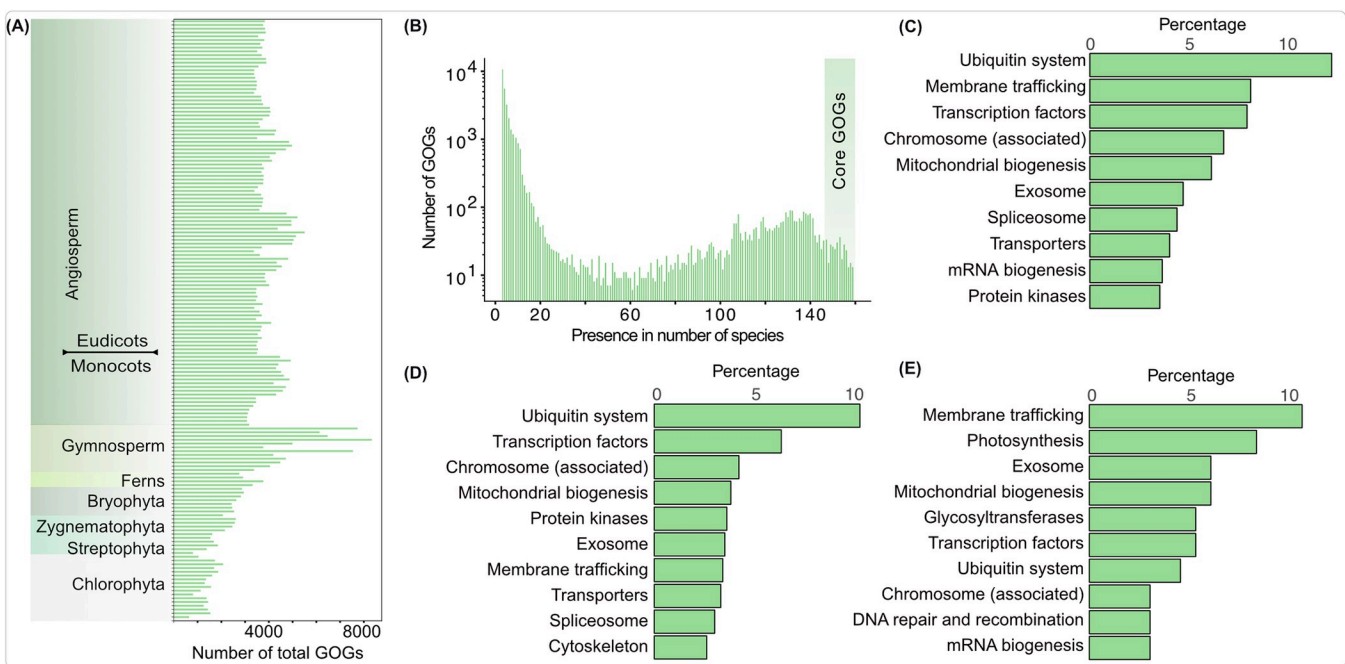

**Fig 1. Distribution and functional annotation of GOGs.** (A) Total number of GOGs present in each species from major Chloroplastida taxa. (B) Number of GOGs as a function of their presence across 159 Chloroplastida species. Major functional categories of 4.71% of all GOGs (c), 3.96% of the embryophyte GOGs (d), and 27.9% of the core GOGs (e). The underlying data of this figure can be found at https://zenodo.org/records/10855592. GOG, green orthogroup.

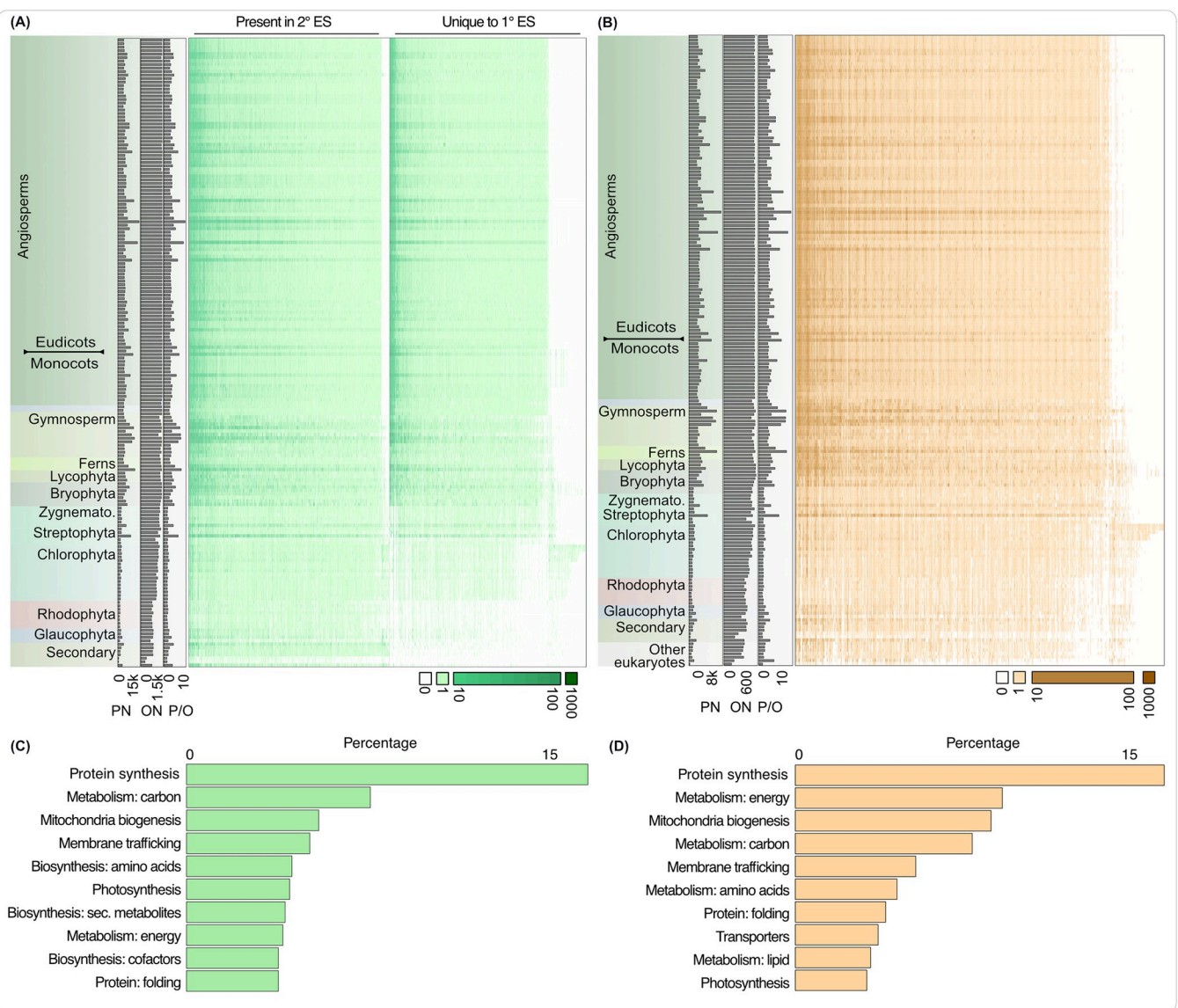

**Fig 2. Mitochondrial and plastid orthogroups across archaeplastidal species.** Distribution of plastid (POGs; A) and mitochondrial orthogroups (MOGs; B). The distribution of POGs was determined for plastids of primary (1° ES) and secondary endosymbiotic origin (2° ES). Protein copy numbers within each POG or MOG across species is shown in the heat-map as per the key on the bottom right of the heatmaps. Horizontal bars on the left side of the heatmaps show the total PNs likely localized to organelles, total POG or MOG numbers (ON) and distribution of PN per OG (P/O) for a given species. Major functional categories of POGs and MOGs in (C) and (D), respectively. The underlying data of this figure can be found at https://zenodo.org/records/10855592. MOG, mitochondrial orthogroup; PN, protein number; POG, plastid orthogroup.

proteome data and homology-based protein clustering of 204 eukaryotes, including that of secondarily photosynthetic eukaryotes (S1B Fig and S1A–S1D Table). In comparison to rhodophytes and glaucophytes, the green lineage encodes almost twice as many POGs (Fig 2A and S1E Table). Within the green lineage, from the Zygnematophyceae and embryophytes onwards, plastid proteomes further expanded both in terms of the number of proteins within each POG and the number of unique POGs. The former is likely a consequence of genome duplications, while the latter underscores functional divergence that followed gene duplications. The distribution of MOGs appears qualitatively similar to that of POGs (Fig 2B and S1F Table). Approximately 60% of the POGs could be functionally annotated, predominantly

operating in biosynthetic and other metabolic pathways such as photosynthesis (Fig 2C and S1G Table). Around 75% of the MOGs could be annotated, containing proteins for mitochondrial biogenesis, membrane trafficking, and translation (Fig 2D and S1H Table). Protein biosynthesis-related proteins are abundant in both, POGs and MOGs, underscoring their biosynthetic activity. Proteins for mitochondrial biogenesis also appear in both. For example, about 60 POGs are annotated as mitochondrial biogenesis. They encompass numerous PPR and mTERF proteins (crucial for RNA editing and metabolism) and proteins involved in various other information processing activities, probable to function in both organelles. Analysis of the N-terminal 20 amino acids show their charge to range from 0 to 2, indicating they might be dually targeted to plastids and mitochondria [42]. Five of the mTERFs are part of a POG and MOG simultaneously (S3D Fig). Overall, the trends show that in embryophytes the number of protein families associated with an endosymbiotic organelle function increased.

The increased number of POGs and MOGs in the green lineage is explained by a combination of 2 phenomena: (a) new gains in the green ancestor; and (b) secondary losses at the origin of rhodophytes [43]. We used ancestral state reconstruction (ASR) to resolve between these 2 possibilities. The branching order of the archaeplastidal lineages remains challenging [44], as sometimes glaucophytes [45] and sometimes rhodophytes come out as the sister to the other remaining archaeplastidal lineages [4,46]. An inferred eukaryotic tree (with 31 non-Archaeplastida eukaryotes as an outgroup to Archaeplastida) placed the rhodophytes and glaucophytes as sister clades (S2 Fig). This tree and the ASR pipeline were validated using rbcS as a control (S3A Fig), and further undergirded the main results which are consistent with varying thresholds of probability of presence and absence in a given ancestor on this eukaryotic tree (S3B Fig), as well as with manually rooting the Archaeplastida to have glaucophytes or rhodophytes as an outgroup to the Chloroplastida (S4–S7 Figs).

The result suggests that the plastid proteome of the last common ancestor of Archaeplastida united ca. 1,000 POGs (Figs 3A, S3B, and S6, S2A–S2C Table). This inferred proteome witnessed significant gains of protein families at the emergence of the green ancestor (and later speciation). Approximately 50% of these newly gained POGs could be functionally annotated (Fig 3C and S3A Table), showing that at the origin of the green lineage novel photosynthesis- and metabolism-related POGs were recruited, while the transition to land (Z/E and embryophyte ancestors) added metabolism-related, as well as protein synthesis- and ubiquitin-related POGs to the toolkit (S3A Table). Using hidden Markov searches, we verify that more than half of the protein families recruited in embryophyte and Z/E ancestors are absent in non-zygnematophyceae algae (S3C Fig). The mitochondrial proteome followed a qualitatively similar trend of expansion (Figs 3B, S3B, and S7, S2D–S2F Table). About 500 MOGs trace back to the archaeplastidal ancestor, while ca. 700 MOGs were identified at the root of angiosperms (Figs 3C and S3B). Around 50% of the newly gained MOGs could be functionally annotated, showing that the chloroplastidal gains contribute to carbon metabolism, protein synthesis, and mitochondrial biogenesis. Terrestrialization also witnessed a similar gain of MOGs, most of which function in metabolism as well as mitochondrial biogenesis and membrane trafficking (Fig 3C and S3B Table).

In summary, across plant species, plastid and mitochondrial proteomes are predicted to have gained a significant number of protein families reflecting the dynamic nature of organellar proteomes post-endosymbiosis [47,48]. A closer look at the function of the newly gained organelle proteins shows a wide variety, including lipid and carbon metabolism, information processing, development, and division of organelles.

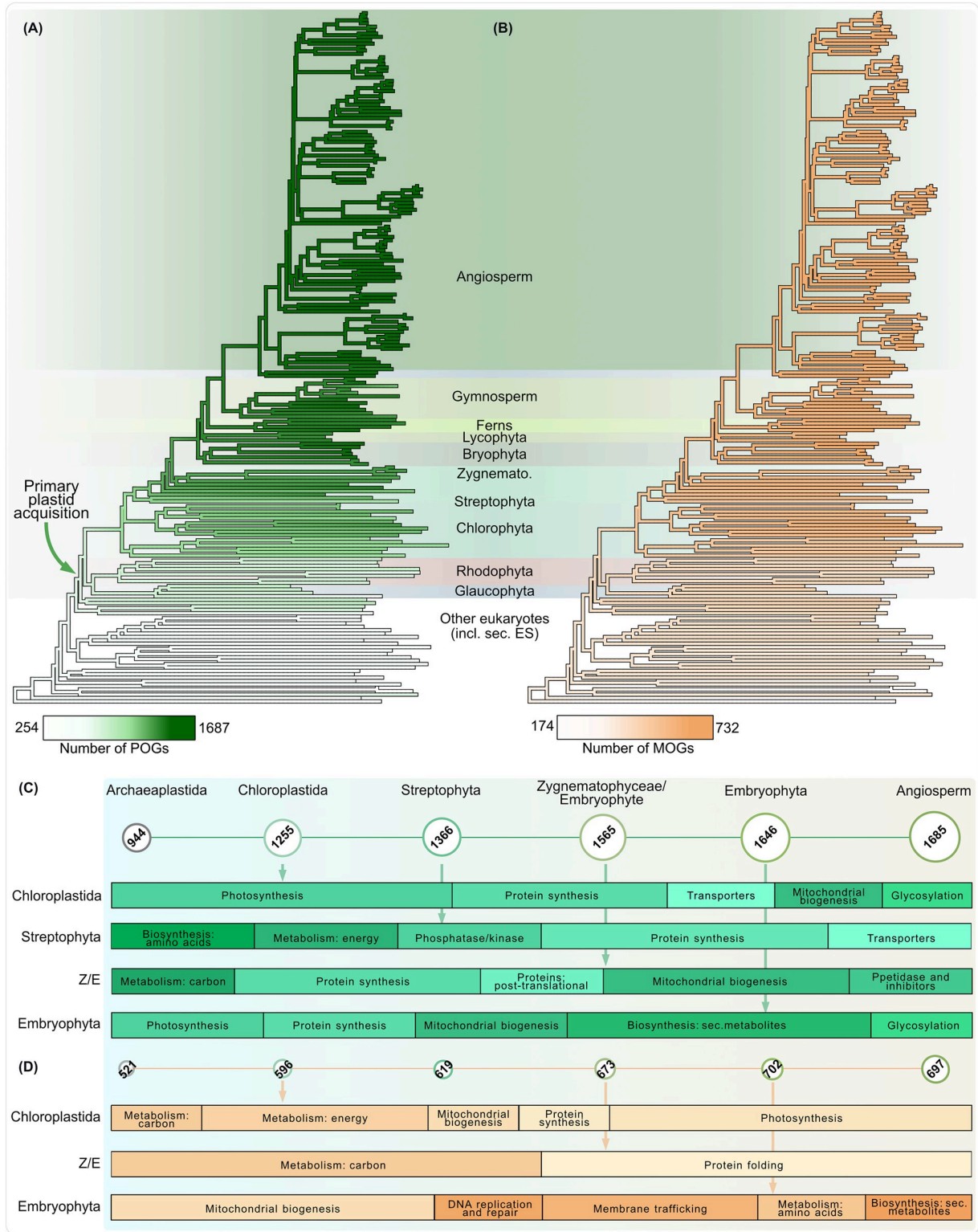

**Fig 3. Evolution of organelle proteomes in Archaeplastida.** Gains in plastid (POGs; A) and mitochondrial orthogroups (MOGs; B) across all nodes of archaeplastidal evolution and POGs coinciding with primary and secondary plastid acquisitions. Gains across main nodes of interest in (C), where each circle represents an ancestor, with its predicted number of protein families shown in the circle and whose diameter correlates with the number of OGs. Major gains occurred in the chloroplastidal ancestor, the common ancestor of Zygnematophyceae and embryophytes (Z/E), and in embryophytes. In (D) the same as in (C), but for mitochondrial OGs. Their functions are shown in the proportionate bar charts

below the ancestors. The underlying data of this figure can be found at https://zenodo.org/records/10855592. MOG, mitochondrial orthogroup; POG, plastid orthogroup.

## Increased complexity of RNA metabolism and photosynthetic adaptability

RNA metabolism such as editing intercepts the linear information flow from mRNA to protein and is crucial for organelles to function [49–51]. Two main domains, the PPR and mTERF domain, are associated with RNA editing and metabolism [52,53]. We first screened for organelle orthogroups containing either of these 2 domains in at least 60% of all proteins within each respective orthogroup (S1C Fig). Around 50 POGs and 20 MOGs were found. More than 80% of them were restricted to embryophytes, only few were present in some algae (Fig 4). A closer look revealed that most of the algal homologues lacked PPR and mTERF domains and they are hence unlikely true orthologues. More generally, this shows that any detailed interpretation regarding an inferred orthogroup's function should be supported by screening for functionally relevant domains.

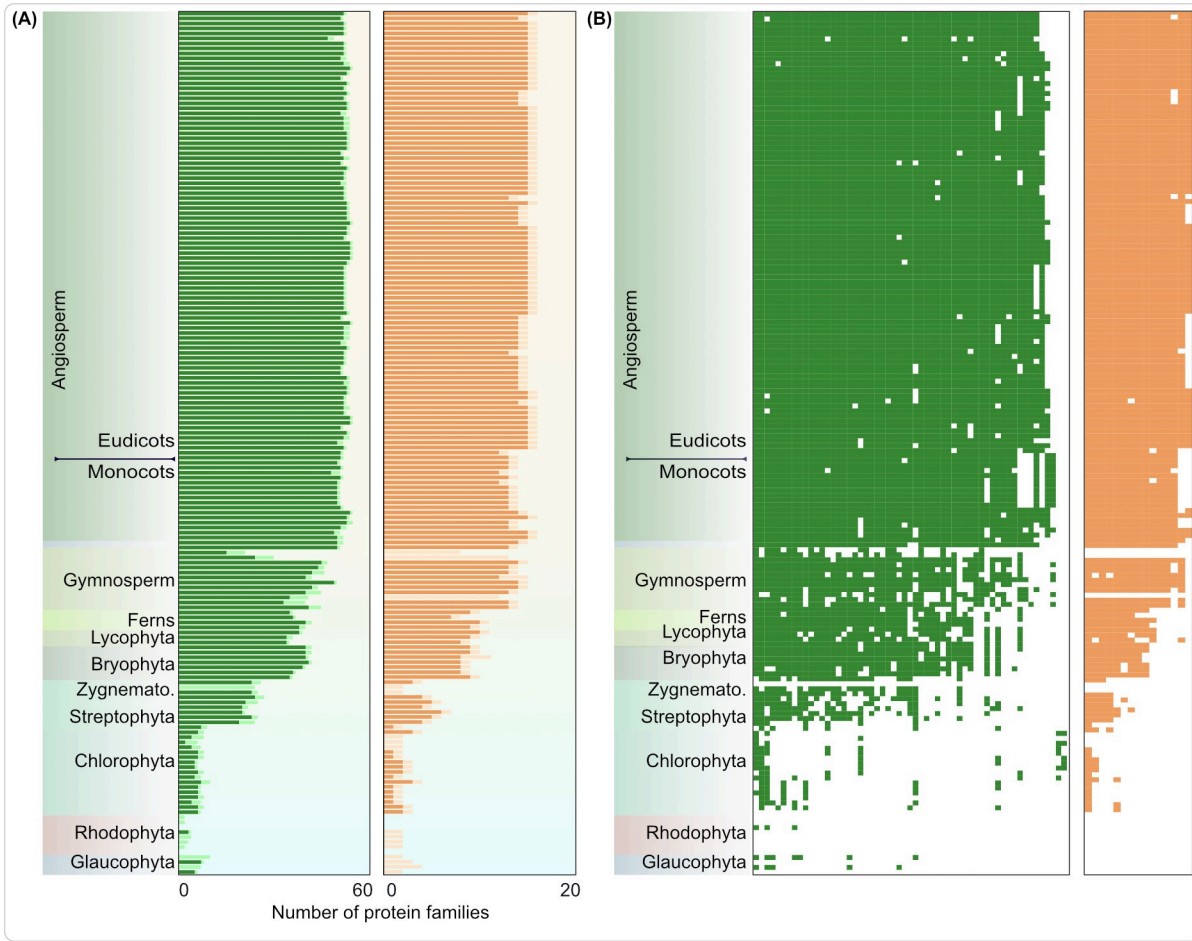

**Fig 4. Recruitment of PPR and mTERF domains in organelle proteins.** (A) Number of POGs (left) and MOGs (right), where at least 1 protein contains a PPR/mTERF domain, is shown in bars with dark shades of colors. Total number of orthogroups (regardless of presence or absence of PPR/mTERF domain in that particular species) is shown in lighter shade. It shows the presence of the orthogroups in question in algae, but that they only later obtained PPR/mTERF domains in embryophytes. (B) Each cell represents an orthogroup and a colored cell indicates the presence of a PPR or mTERF domain in the protein family (column) of a respective species (rows). The underlying data of this figure can be found at https://zenodo.org/records/10855592. MOG, mitochondrial orthogroup; POG, plastid orthogroup.

True PPR or mTERF domain-containing RNA-editing proteins (and splicing and processing at large) increased significantly in number by recruiting new orthogroups, also through adding the 2 domains to proteins that did not contain these in their algal ancestor. A presence–absence pattern shows that >90% of proteins containing PPR/mTERF domains are exclusive to land plants, except for *Chara braunii* and *Klebsormidium flaccidum* (Fig 4B). These proteins include, but are not limited to, OTP51 and SOT5 (present in embryophytes and *Chara*) as well as SOT1, SVR7, THA8, PDM4 (present only in embryophytes; S9 Fig). Target transcripts of these RNA metabolism factors point to the synthesis and assembly of photosynthesis-related proteins and to proteins of the thylakoid membrane. Likewise, mTERFs, which are crucial for plastid and leaf development, are also uniquely expanded in the terrestrial clade with examples of protein re-targeting across organelles [54]. The dual targeted (plastid and mitochondrion) mTERF6, unique to the land plants (S9 Fig) and the streptophyte alga *Klebsormidium*, takes part in retrograde signaling to the nucleus via ABA and imparts abiotic stress tolerance [55]. Overall, RNA metabolism across plants has undergone major changes and has a significant impact on photosynthesis, improvement of which was key to thriving on land.

## Adaptation to the terrestrial habitat and changes in plastid biochemistry

Main terrestrial stresses include draught, high (UV-)light, and swift temperature changes. Cutin and suberin, 2 of the most abundant lipid polymers on Earth [56], evolved as one countermeasure [9]. We find that cutin and suberin evolution was enabled by the recruitment of an organelle-specific GPAT (glycerol-3-phosphate acyltransferases) family in the embryophyte ancestor (Fig 5), which includes GPAT1 (mitochondrial), GPAT 4,6 and 8 of the endoplasmic reticulum [57,58]. Trafficking of these lipids across organelles was made possible by a dual targeted TGD4 [59] that was recruited in the chloroplastida ancestor (Fig 5). Acyl carrier thioesterases, responsible for the export of fatty acids from the plastid, acyl carrier protein desaturases (ACP-desaturase), and acyl-carrier proteins co-factors of fatty acid bio-synthesis were uniquely retained and expanded in the green lineage (S9 Fig). Duplication and divergence of ACP desaturases in embryo- and spermatophytes played an important role in regulating lipid composition shifts in response to temperature and drought, the regulation of seed oil content and development [60]. Likewise, acyl-carrier proteins also increased in copy number (S9 Fig) and adapted towards a light-induced expression and regulation of the seed fatty acid content [61,62]. These changes in organelle lipid synthesis and trafficking underpinned embryophyte adaptations to cope with draught and high temperature stress (wax biosynthesis, deposition on the layer of leaves, and cuticle development), as well as seed development and germination in spermatophytes (Fig 6D).

Changes in starch metabolism mostly pertain to its regulation. ADP-glucose pyrophoshorylase (AGPase), an enzyme responsible for a rate-limiting step in starch metabolism, is uniquely retained in the green lineage and increased in copy number in streptophytes (S9 Fig). AGPases diverged to regulate starch metabolism under osmotic and light stress, as well as the differential regulation of starch synthesis and degradation [63–67]. Another key regulatory enzyme, PGI (phosphoglucose isomerase) evolved a distinct family (PGI1) in Zygnematophyceae (S9 Fig). It likely kickstarted the regulation of starch metabolism at the water-to-land interface and later assumed significant roles in embryophyte fatty acid content regulation and the yield of seeds [68]. PTST3 also emerged around the time of terrestrialization (S9 Fig), which evolved to regulate starch synthesis with significant impact on plastid development [69]. In contrast to the flow of carbon through glycolysis, GSM2 (which originated in streptophytes; S9 Fig), shunts carbon towards the pentose-phosphate pathway and protects plastids from oxidative stress in *Arabidopsis* [70].

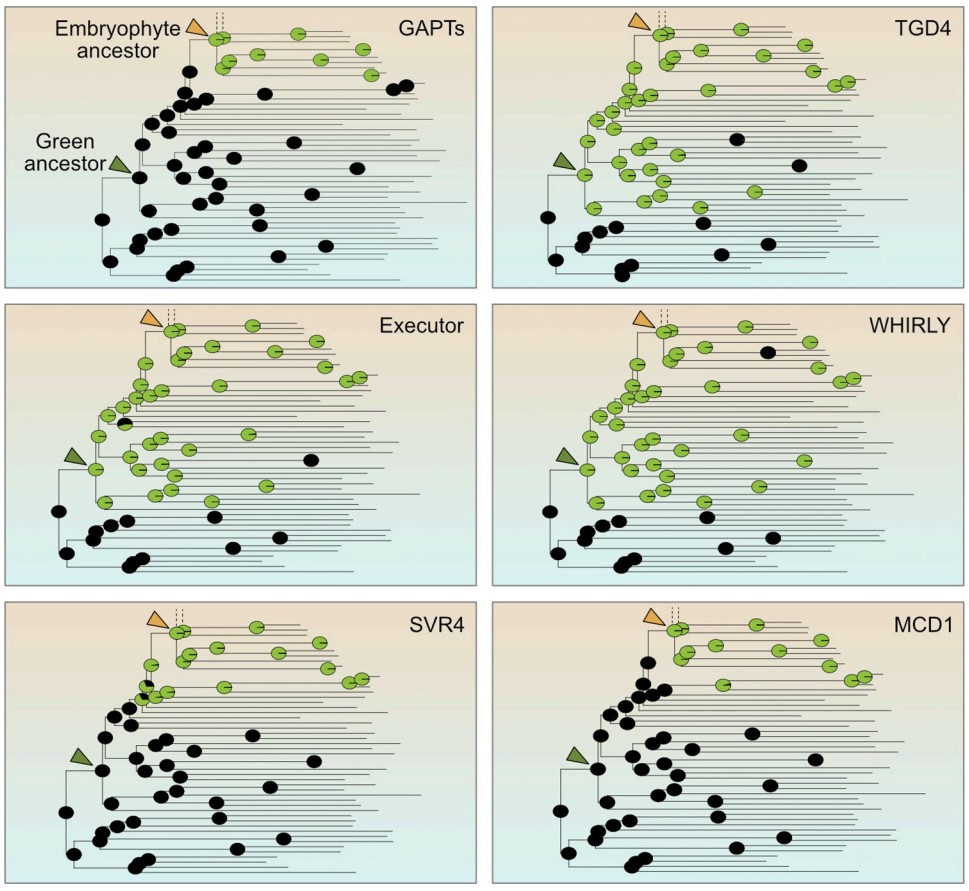

**Fig 5. Origins of key proteins involved in metabolism, communication, and development.** ASR for selected lipid metabolism (GAPT and TGD4), retrograde signaling (Executor and Whirly), plastid development (SVR4), and division (MCD1)-related proteins. The pie charts at each node represent the probability of presence (green) or absence (black) of a protein family in that node. The underlying data of this figure can be found at https://zenodo.org/records/10855592. ASR, ancestral state reconstruction.

## Emergence of sophisticated antero- and retrograde communication cascades

Communication across compartments is critical for a concerted response to environmental stimuli. Plastids are key environmental sensors that interconnect cellular metabolism with physiological requirements and stress responses, and terrestrial stressors are key triggers of plastid-to-nucleus retrograde signaling [12,13,22]. We screened for the origin and diversification of EXECUTOR and SVR4, both components of retrograde signaling. We also screened for WHIRLY, a protein family that acts on RNA splicing and ribosome biogenesis, but also relocates between compartments and remains a disputed candidate for retrograde signaling [18,71–75]. EXECUTOR, key to regulating retrograde signaling, oxygen and light stress regulation [76–78], originated in the ancestor of the Chloroplastida and so did WHIRLY (Fig 5); the latter underwent copy number expansion in embryophytes and was likely lost in some bryophytes (S9 Fig). Divergence of these copies led to a localization across multiple organelles and today they are crucial for maintaining functional respiration, photosynthesis, and the response of mitochondria and plastids to biotic and abiotic stresses [79–81]. Additional

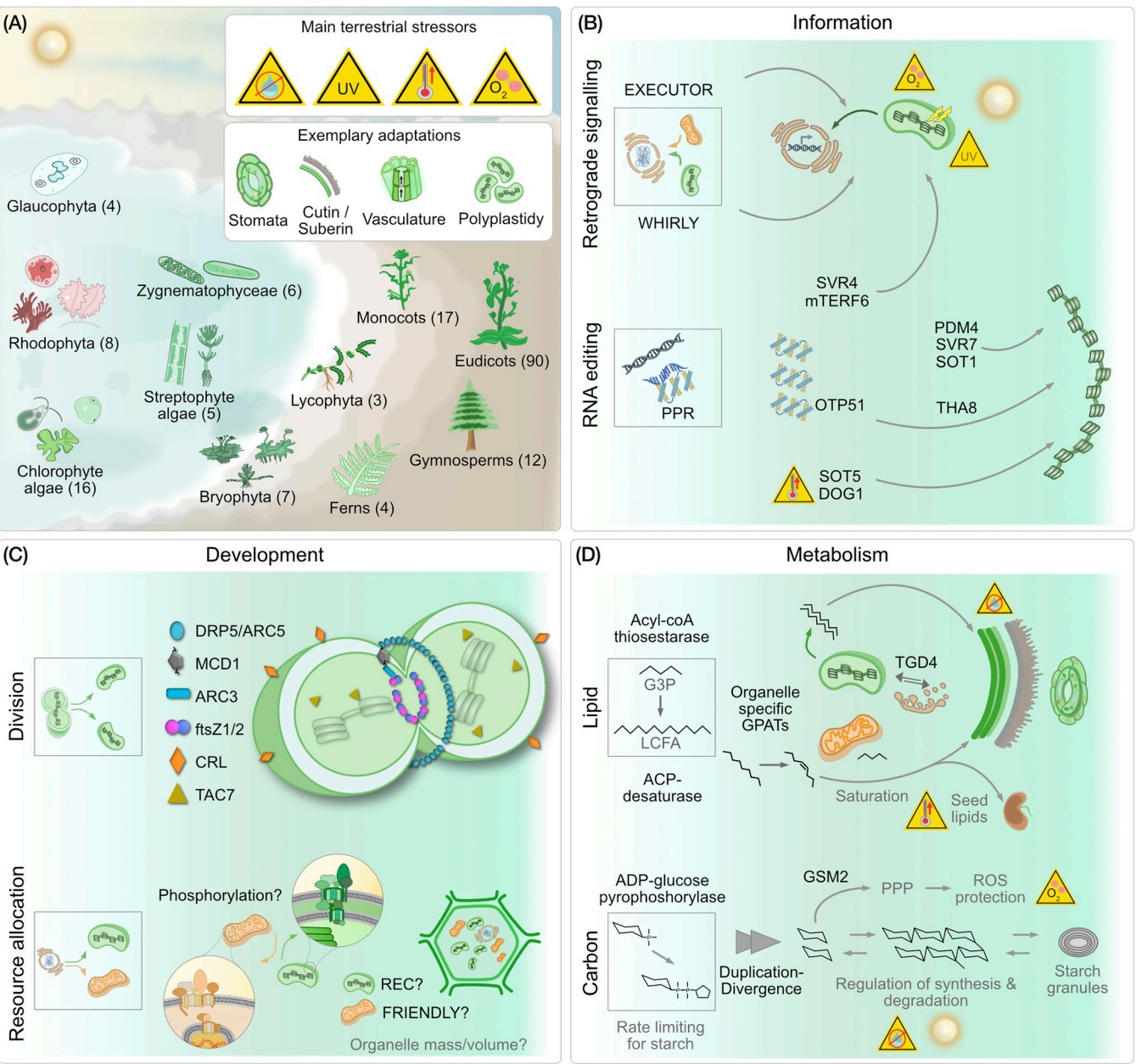

**Fig 6. The global greening and endosymbiotic organelles.** (A) After the endosymbiotic origin of the plastid, 3 aboriginal lineages emerged that form the Archaeplastida: the glaucophytes, rhodophytes, and chlorophytes. From the latter, streptophyte algae evolved, including the zygnematophyceae, that represent the algal sister clade to land plants (embryophytes). Abiotic stresses encountered during terrestrialization (water scarcity, high UV, swiftly altering temperatures and higher levels of $O_2$) selected for adaptive features such as stomata and a cutin layer. The numbers in parenthesis indicate the number of genomes from each major group that was screened. Recruitment of new organelle proteins improved 3 key aspects of organelle biology in light of terrestrialization: (B) information processing, (C) development, and (D) metabolism. Details for each tile are discussed in the main text. UV, ultraviolet.

paralogs evolved, each with a specific function in the main green lineages, and they likely aided in the colonization of the terrestrial habitat by the ancestor of land plants (Fig 6B).

SVR4, a dual targeted (plastid and nucleus) protein recruited during the time of terrestrialization (Fig 5), likely communicates required gene expression changes needed for light-induced plastid development, thylakoid stacking, and thermomorphogenesis [82,83]. In combination,

this facilitates light-induced photomorphogenesis, a process key for surviving on land. An increase in the complexity of retrograde signaling was a precursor for terrestrialization [12], for instance via innovations associated with the the 3′-phosphoadenosine-5′-phosphate family, which promoted the emergence of stomatal closing in land plants [84]. The recruitment and diversification of the proteins we highlight were quintessential for responding to 2 major stressors that are more pronounced and more rapidly changing on land than in water: light and temperature (Fig 6B).

## Recruitment of new proteins and changes in organelle development

The coordination of tissue and plastid development is linked to ensure an appropriate response to biotic and abiotic factors, especially in morphologically complex plants [85–87]. Polyplastidy is a trait of land plants and some macroscopic algae such as *Bryopsis* or *Chara* [15], and known molecular determinants include MinD, MinE, ARC3, and the FtsZ proteins [16,86]. Our data supports that MULTIPLE CHLOROPLAST DIVISION SITE 1 (MCD1), another known component of the plastid division machinery [88], originated in the ancestral embryophyte (Fig 5). The cotyledon chloroplast biogenesis factor CYO1 and the transcription-ally active chromosome factor 7 (TAC7) are important components of thylakoid biogenesis and the plastid translation machinery, respectively. Both originated in the streptophyte ances-tor (S9 Fig) and play key roles in chloroplast, cotyledon, thylakoid and leaf development in *Arabidopsis* [89–91]. Lastly, CRUMPLED LEAF (CRL), a protein residing in the outer plastid membrane, emerged during terrestrialization, too (S9 Fig), likely for regulating plastid division and securing correct plastid inheritance during embryogenesis [92,93].

Crucial for plastid biogenesis, especially in light of an expanding proteome, is the import of proteins. The membrane GTPase TOC159 is essential for chloroplast biogenesis via the selec-tive recognition and import of the photosynthetic proteins [94] and is unique to the green line-age (S9 Fig). The membrane recruitment of this protein requires TOC75, of which a special variant evolved in the green ancestor after the duplication of OEP80 [14,95]. The copy number of TOC159 expanded from the Zygnematophyceae onwards (S9 Fig), hinting at its functional diversification. Unlike in the chlorophyte alga *Chlamydomonas*, land plant TOC159 homo-logues possess an N-terminal acidic domain that gets phosphorylated to alter substrate speci-ficity [94,96]. Furthermore, TOC159, along with TOC132 and TOC120, play important roles in regulating plastid lipid synthesis and membrane fluidity and in *Arabidopsis* show tissue-spe-cific expression (The Arabidopsis Information Resource) [97–99]. Further on the course of evolution, the J-domain-containing protein TOC12 [100] was likely recruited in the ancestral embryophyte for supporting the import machinery at the intermembrane space (S9 Fig). The terrestrial habitat demands a highly efficient and fluid import of proteins, for example, upon high light and other abiotic stresses [14,101]. The expansion of the TOC/TIC system in the embryophyte ancestor reflects how the organelle dealt with an ever-increasing diversity of sub-strates that were required to be processed.

## Discussion

The settling of land by a streptophyte alga and the subsequent evolution and spreading of plants (Fig 6A) was pivotal in the transformation of the terrestrial habitat and it laid the foun-dation for the concomitant evolution and diversification of animals [1,2]. Throughout the hundreds of millions of years of plant evolution, both organelles of endosymbiotic origin underwent a multitude of molecular adaptations, hereby evolving into the plastid and mito-chondrion of modern plants. We identified 31,650 protein families unique to the green lineage, approximately 50% of which are unique to embryophytes. It demonstrates an expansion and

divergence of protein families at the time of plant terrestrialization and in line with a recent study that identified around 10,000 duplications at the birth of embryophytes [102].

Expansion of proteins families is evident in both organellar proteomes at the origin of the green lineage itself and at the water-to-land transition. The gain of protein families at the origin of the Chloroplastida needs to be treated with caution due to the documented genetic bottleneck that characterizes rhodophyte origin [103–107] and the sparse availability of glaucophyte genome data. Some of the newly recruited protein families at the origin of the green lineage might rather be explained by a loss in rhodophytes and a retention in the chloroplastidal ancestor instead of a gain. Regardless, this has little bearing on the biological significance of a given protein family with respect to the overall increase in complexity of organelle biology—both concerning the variety as well as the number of proteins targeted to plastids and mitochondria—throughout streptophyte evolution. It affected the organelles metabolic, informational and developmental complexity, and facilitated the evolutionary successful transition from water to land more than 500 million years ago (Fig 6).

Changes in organelle lipid biochemistry contributed to one of the key adaptations in land plants that is the cuticle. Land plant GPATs (crucial to lipid synthesis for cutin and suberin) contribute to increased hydrophobicity and water retention in embryophytes [9] and their activity in embryophytes differ from that in algae [108,109]. Our analyses pinpoint the origins of organelle-specific GPATs (GPAT 1,4,6, and 8) to the embryophyte ancestor, and of which deleting GPAT4 and GPAT8 distorts cuticles and increases water loss by several fold [57,58]. In parallel, lipid trafficking was mediated by the recruitment or divergence of proteins such as TGD4 and acyl carrier thioesterases, which contributed to wax biosynthesis and deposition on leaves, cuticle development, thylakoid membrane stacking [59], seed development, and germination [60]. As for starch metabolism, the archaeplastidal ancestor likely stored starch in the cytosol [110], but the red and green lineage experienced different fates from there on. Rhodophytes continued to store starch in the cytosol in the form of Floridean starch [111], while in the green lineage, particularly in complex plants, more localized control of starch synthesis and degradation was facilitated by changes in regulatory proteins (e.g., AGPase). Together, organelle metabolism evolved to serve key roles in the synthesis, regulation, and trafficking of lipids involved in wax coating to prevent water loss in the land plant ancestor, as well as synthesis and storage of starch (Fig 6D).

RNA processing and editing is a crucial component of information processing and overall functionality of plant organelles [49,50]. Changes in RNA metabolism are evident at the origin of the green lineage, where RNase-P (tRNA maturation) was replaced by Protein-only RNase P or PROPs [112,113]. Subsequent expansion of PROPs in embryophytes (S9 Fig) led to organelle-localized copies, of which some are essential for maintaining organelle morphology, function, and plant viability [114]. Components associated with plastid-encoded RNA polymerase (PEP associated proteins, PAPs) also show a gradual recruitment from the green ancestor to embryophyte ancestor (S8 Fig). RNA editing of C to U is not found in algae, however, and editing sites in embryophytes are unlike those of any other eukaryote, suggesting they emerged independently [50]. We trace the emergence of many RNA-metabolism proteins to the time of plant terrestrialization and their known targets are transcripts involved in photosynthesis and stress tolerance-related transcripts, both key to colonizing land (Fig 6B). For example, THA8, PDM4, SVR7, and SOT1 associate with transcripts such as ycf2 and ycf3, and contribute to thylakoid development and biogenesis [115,116], the generation of photosynthetic complex proteins, grana stacking, and embryo and plastid development [115,117,118]. OTP51 and SOT5 splice transcripts related to chlorophyll synthesis, photosynthesis, and thylakoid membranes (ycf3, TRNK, and RPL2) [119–121], whereas DOG1 is important for high temperature response and chloroplast development [122]. This elaborate RNA processing in organelles,

especially plastids, appears to serve photosynthesis (and thylakoid)-related transcripts. It is feasible that by benefitting photosynthesis, organelle RNA editing continued to be positively selected for during terrestrialization and was expanded.

One evolutionary step towards efficient photosynthesis, where RNA editing also plays a key role, are grana stacks [85]. The evolutionary origin of grana remains elusive, along with the underlying developmental pathways involved in regulating its formation and maintenance [85,123,124]. Highly organized grana stacks are observed in embryophytes and some Zygnematophyceae (e.g., the *Cosmarium* genus) [125], but not chlorophytes such as *Chlamydomonas* [126]. We noticed a patchy distribution of grana morphology-associated proteins such as CURT1, RIQ1, and RIQ2 (S9 Fig), with both RIQs being present in all streptophytes and some chlorophytes but excluding *Chlamydomonas*. In light of the many key adaptations in Zygnematophyceae discussed here and elsewhere [11,127], we speculate that a sophisticated stacking of grana originated in streptophytes and was beneficial for thriving on land through photosynthesis optimization, in particular, with respect to photosystem repair and the separation of the photosystems and the ATP synthase [128,129].

This expansion of an organelle proteome necessitates improving the capacity to import proteins. Changes in some import receptors within the green lineage and in targeting sequences at its origins are known, with phosphorylation likely emerging as a key regulator for sorting the newly expanded proteome differentially to plastid and mitochondria (Fig 6C) [14,130]. Despite such adaptations, protein sorting is never perfect and some mistargeting might be positively selected for. A regulated distribution of newly recruited proteins (e.g., WHIRLY, TGD4, mTERF6; Fig 6B) to multiple organelles (with distinct organellar functions) hints at adaptive values of this apparent mis-sorting. How many of the newly recruited proteins get "mis-sorted" owing to biological adaptability versus stochasticity remains to be explored together with obtaining a more comprehensive picture of (regulatory) mechanisms associated with sorting in general.

Embryophyte cells target proteins not to a single plastid, but many simultaneously. The presence of multiple plastids per cell, (polyplastidy), in the green lineage, evolved in an embryophyte ancestor, maybe the common ancestor of embryo–and charophytes, and through changes in plastid fission and a decoupling of the latter from the cell cycle [15,16]. We find that MCD1, a core regulator of the plastid division proteins FtsZ2 and ARC3, emerged in the embryophyte ancestor, which corroborates the idea of a mono- to polyplastidy switch during the transition from water to land [12,15,16,131]. A change in the copy number of plastids also requires a mechanism that maintains a functional organelle to cell volume ratio and resource allocation (Fig 6C). The REDUCED CHLOROPLAST COVERAGE (REC) protein is involved in such a mechanism in *Arabidopsis* [132] and the phylogenetically related protein FRIENDLY regulates the distribution of mitochondria, also in plants and non-photosynthetic organisms [133,134]. REC and FRIENDLY share almost all of their domains. How they exactly function and differentiate between the 2 organelles remains elusive. From what we can tell, FRIENDLY emerged during eukaryogenesis and the origin of mitochondria. REC we can trace back to the streptophyte ancestor (S9 Fig) and after a duplication event of FRIENDLY. We speculate that the origin of REC helped to cement polyplastidy, which itself supports larger body plans and the diversification of different plastid types [15]. Lastly, an increase in organelle copy number also requires an overall increase in the capacity to synthesize proteins. The largest fraction of organelle proteins operate in tRNA, amino acid, and ribosomal biosynthesis and undergird the biosynthetic capacity of organelles, an adaptation strategy akin to their bacterial ancestor [135,136].

The accommodation of the early mitochondrial endosymbiont is associated with the origin of the endomembrane system and necessitated the emergence of eukaryotic traits including

mito- and autophagy [137–139]. Our analyses show that the integration of a subsequent endo-symbiont, the plastid, coincided with the emergence of proteins that work for the endomem-brane system. Salient are changes in the ubiquitin system during terrestrialization, when polyplastidy in the green lineage also emerged (S1G Table). Ubiquitination is key to proteo-some-mediated degradation and is performed chiefly by the E3 ubiquitin ligase family, which are important in land plants also for photomorphogenesis [140]. RING (really interesting new gene) E3 ligases contribute to growth, development, and stress response via also mediating protein–protein interactions [141–144]. We trace a number of RING finger and related pro-teins to terrestrialization (S9 Fig) that include, but are not limited to, *DAL1* and *DAL2* (for *Drosophila* DIAP1 like 1 and 2), KEG (Keep on going), and NIP1 and NIP2. *DAL1* and *DAL2* play a key role in regulation of programmed cell death [145], peroxisome, and chloroplast bio-genesis [146–148]. KEG contributes to stress mitigation [149,150], while NIP1 and NIP2 play a role in plastid development by docking plastid RNA polymerase to the thylakoid membrane [151]. The regulated degradation of plastids and other changes in the endomembrane system are a prerequisite for housing multiple plastids per cell and we find many more recruitments broadly affiliated with the endomembrane system, with poorly characterized functions until now. Exploring the functions of these proteins will add valuable insights into the cell biological changes that endosymbiosis stipulates.

While we focus on the evolution of the chloroplast of the green lineage, the rhodoplast of rhodophytes and the cyanelle of glaucophytes, as well as the plastid of secondary plastids out-side of the Archaeplastida are worth considering. Based on the available chloroplastidal and glaucophyte data, our analysis predicts about 900 plastid proteins for rhodophytes (Fig 2A). This number is likely skewed by the sequence divergence of over a billion years, however, and even relaxed homology searches—e.g., on the basis of hidden Markov models—are no replace-ment for experimental validation (S3D Fig). As for plastids of secondary origin, we find that around half of the plastid proteins, or fewer, are shared between primary and secondary plas-tids (Fig 2A). Experimental proteomes of secondary plastids are available for *Phaeodactylum* [152] and *Plasmodium falciparum* [153], and the numbers match ours quite well (839 versus 934, and 346 versus 329, respectively; Fig 2A). Still, the experimentally validated proteins of secondary plastids remain entangled with chloroplastidal proteomes [152], as the filtering involves matches to just those. The experimental data, as well as our predictions are likely underestimating the true nature of organelle proteomes to a degree due to sequence diver-gence and a niche-specific trajectory of primary and secondary plastids. This further under-scores the need for additional experimental proteomes from species other than chloroplastida, and which could benefit from the use of subcellular localization mapping using LOPIT [157] and proximity labeling [158].

In closing, although experimentally reported plant plastid and mitochondrial proteomes are scarce, we were able to generate a first comprehensive molecular atlas of the changes of plastid and mitochondrial protein families in the evolution of the green lineage. ASR allows to map the organelle transformations that facilitated the major transitions such as terrestrializa-tion and which will improve with every new proteome that is added. By inferring plastid and mitochondrial proteomes for 173 species, we set testable expectations for new proteomes to come and provide a solid database, where origins and across species orthologues of any known (organelle) protein can be searched (S1C and S1D Table). Additional proteomes, once avail-able, will likely solidify the general pattern observed and uncover more lineage-specific curiosi-ties. We identify numerous mitochondrial protein recruitments, whose physiological roles and adaptive values help to better understand plant mitochondrial biology. For plastid proteins, we infer their functions and physiological importance based on the extensively studied *Arabidop-sis* system. Utilizing an advanced orthology search technique [40], we postulate that

orthologues of *Arabidopsis* are likely to exhibit similar functions in other species. Our methodologically robust approach maps various changes in evolution associated in particular with terrestrialization that can now be experimentally explored across selected models and with a focus on less-well studied streptophyte algal and bryophyte species [156,157].

## Conclusions

Endosymbiotic organelles have a distinct place in the evolutionary tapestry of life. Through the combination of organelle proteome data and phylogeny, we trace the evolution of mitochondria and plastids over a span of a billion years of plant evolution by inferring their proteomes for over a hundred Archaeplastida species. Our comprehensive molecular atlas identifies main changes in their metabolism, communication, information processing, and biogenesis. Key adaptations in plant organelles fostered the emergence of wax and cutin (see organelle lipid synthesis and transport), improved the photosynthetic yield (see organelle RNA metabolism and highly structured grana stacks), and the response to abiotic stressors (see inter-organelle communication), and mediated the transition from mono- to polyplastidy (see division and volume control). By connecting the molecular adaptations of mitochondria and plastids to macroevolutionary trends, we show how important changes in organelles of endosymbiotic origin were for the speciation that gave rise to the Chloroplastida and later the origin of land plants from a charophyte algal ancestor.

## Material and methods

### Curation of green orthogroups (GOGs)

Input protein sequences from 686 proteomes (from KEGG [158] and Phytozome [29] were clustered using Orthofinder version 2.5.4 [40], after all versus all blasts were conducted (E-value cutoff 10e-10) using diamond blast version2.011 [38]. From orthogroups (OGs) recovered, OGs with at least 3 Chloroplastida species and less than 3 species other than Chloroplastida were annotated as green orthogroup (GOGs). Schematic in S1A Fig. Inhouse python script used for this, the resulting data and other data processing, are available at https://zenodo.org/records/10855592 [42].

### Curation of plastid and mitochondria orthogroups (POGs and MOGs)

A total of 5,452,977 proteins from 204 eukaryotes (S1A Table) were clustered using Orthofinder as described above. Orthogroups that contained at least 1 experimentally verified organelle protein from any one of the 4 experimentally verified organelle proteome of *C. reinhardtii* [159], *P. patens* [160], *Z. mays* [161], *A. thaliana* [161], were annotated as organelle (plastid and mitochondria) orthogroups. Schematic in S1B Fig.

### Functional annotation of orthogroups

The source of >90% species was Kyoto Encyclopedia of Genes and Genomes (KEGG), which included KEGG orthology identification (KOID) for protein sequences. For all proteins within each GOG, KOIDs were retrieved and the most frequent KOID (i.e., majority rule) was annotated to each GOG (S1C Fig). From the assigned KOIDs, their KO BRITE functional category was assigned to each GOG. KOIDs for POGs and MOGs were retrieved the same way. For each KOID, the pathway names and BRITE categories at various level of resolutions were used for assigning functional categories manually to each OG. Manual assignment was necessary since BRITE names included a large fraction of categories such as "enzymes" and "exosomes." These were either not very informative or were misleading as many of "exosome" annotated

proteins took part in protein synthesis or folding. Lastly, for OGs or proteins discussed with respect to their physiological relevance, the functions were retrieved from the literature (cited in the text).

### Inference of ancestral states

A phylogeny of Archaeplastidal species was inferred based on all genes conserved in all species, using "Species tree inference from all genes (STAG)" method [162], as a part of orthofinder analysis. STAG infers a species tree by taking greedy consensus of gene trees from each protein family (including that of multigene families). This phylogeny was rooted using minimal ancestral deviation [163] which places Rhodophyta as the sister to all others. Independently, the same unrooted phylogeny was manually rooted using FigTree (v1.4.4) [164] such that Glaucophyta were at the base. Ancestor state of presence and absence of organelle protein families across nodes were inferred using Phytool [165] package 0.7.80. Based on character state at the tips of the tree, Phytool inferred Bayesian posterior probabilities under a single rate model [166,167] of the character state across nodes of the tree. All Ogs that were present in major ancestors of plant groups with probability higher than 0.75 and absent in the preceding ancestor were considered as newly recruited in that lineage. Ogs or proteins discussed with respect to its physiological role in a given clade, their absence outside the group was verified in our copy number database as well as on homologue database available on TAIR.

### Searching for potential RNA metabolism POGs and MOGs

Hidden Markov models (HMM) of PPR and mTERF domains were downloaded from pFAM [168] with the IDs: PF01535, PF12854, PF13041, PF13812, and PF02536. Each of these HMMS was used as a query to search against the full sequences of all proteins within each POG and MOG. If a given OG had more than 60% of individual proteins containing PPR or mTERF, the OG was annotated as RNA metabolism OG. Origin of such Ogs were traced using ASR as described above.

### Supporting information

**S1 Fig. Sorting and annotating orthogroups.** Sorting of **(A)** the green orthogroups (GOGs) and **(B)** plastid orthogroups (POGs). In the first step (top boxes), source protein sequences from available species were clustered into protein families. Based on predetermined criteria (between the 2 boxes), the protein clusters were then separated into the clusters of interest (bottom boxes). Mitochondrial orthogroups (MOGs) were sorted the same as **(B)** and based on their presence in any experimental mitochondrial proteome. **(C)** For the functional annotations (for GOGs, POGs, and MOGs), KEGG KOID were translated from ProteinID of each protein present in each cluster and from across species. For species outside of the KEGG database (or some proteins within the KEGG database), no KOIDs are available, they are indicated by "N/A." The most frequent KOID within a cluster (e.g., KEGG ID 1 in the second circle) was assigned as the KOID for the entire cluster. **(D)** Ogs were sorted into PPR or mTERF containing Ogs by using the hidden Markov profiles of these domains as a query against all proteins in a given OG. If more than 60% of individual proteins inside an OG contained these domains, we labeled it as RNA editing domain.
(TIFF)

**S2 Fig. Phylogeny of 204 eukaryotes.** Inferred phylogeny of 204 eukaryotes, with major groups and ancestors of Archaeplastida indicated. The underlying data of this figure can be

found at https://zenodo.org/records/10855592.
(TIFF)

**S3 Fig. Validation of ancestor state reconstruction (ASR).** Validation of ancestor state reconstruction (ASR) approach on a control protein family of rbcS **(A)**. Number of POGs and MOGs gained by major ancestors, as per probability threshold of inclusion 0.65, 0.75, and 0.85 **(B)**. Hidden Markov model-based validation of newly gained POGs of Z/E and Embryophyte ancestor **(C)**. Charge of the first 20 amino acids across mitochondrial biogenesis related POGs **(D)**, with mTERF containing POGs present also in MOG shown in a darker shade. The underlying data of this figure can be found at https://zenodo.org/records/10855592.
(TIFF)

**S4 Fig. Phylogeny of Archaeplastida with rhodophyta at the base.** Inferred phylogeny of Archaeplastida (rhodophytes as the sister lineage to all others) with major ancestor nodes indicated with the arrows and major groups highlighted by labels on the right. The underlying data of this figure can be found at https://zenodo.org/records/10855592.
(TIFF)

**S5 Fig. Phylogeny of Archaeplastida with laucophyte at the base.** Inferred phylogeny of Archaeplastida (glaucophytes as the sister lineage to all others) with major ancestor nodes indicated with the arrows and major groups highlighted and labeled on the right side. The underlying data of this figure can be found at https://zenodo.org/records/10855592.
(TIFF)

**S6 Fig. The evolution of POGs.** Evolution of plastid orthogroup numbers across the Archaeplastida on a phylogeny with Rhodophyta **(A)** and Glaucophyta **(B)** as a basal branch. The underlying data of this figure can be found at https://zenodo.org/records/10855592.
(TIFF)

**S7 Fig. The evolution of MOGs.** Evolution of Mitochondrial orthogroup numbers across the Archaeplastida on a phylogeny with Rhodophyta **(A)** and Glaucophyta **(B)** as a basal branch. The underlying data of this figure can be found at https://zenodo.org/records/10855592.
(TIFF)

**S8 Fig. ASR for RNA polymerase interacting proteins.** Ancestor state reconstruction (ASR) **(A)** and gene copy numbers **(B)** for selected plastid encoded RNA polymerase interacting proteins (PAPs). The pie charts at each node represent the probability of presence (green) or absence (black) of a protein family in that node. The underlying data of this figure can be found at https://zenodo.org/records/10855592.
(TIFF)

**S9 Fig. Copy number distribution of key proteins recruited around terrestrialization (white indicates absence of a protein, values above 50 were shown as the darkest shade).** The underlying data of this figure can be found at https://zenodo.org/records/10855592.
(TIFF)

**S1 Table. Clustering to identify plastid and mitochondria orthogroups (POGs and MOGs).**
(XLSX)

**S2 Table. Presence-absence of each orthogroup across all ancestors.**
(XLSX)

**S3 Table. Functions of newly recruited orthogroups. All available at:** https://zenodo.org/records/10855592 [42].
(XLSX)

## Acknowledgments

We acknowledge support from the high-performance computing cluster (HILBERT) at the Heinrich Heine University Düsseldorf and thank Michael Knopp (HHU Düsseldorf) for his help. We also thank Alice Barkan (University of Oregon) for her useful feedback on our bioRxiv preprint.

## Author Contributions

**Conceptualization:** Parth K. Raval, Sven B. Gould.

**Data curation:** Parth K. Raval.

**Formal analysis:** Parth K. Raval, Alexander I. MacLeod.

**Funding acquisition:** Sven B. Gould.

**Investigation:** Parth K. Raval, Alexander I. MacLeod.

**Methodology:** Parth K. Raval, Alexander I. MacLeod.

**Project administration:** Sven B. Gould.

**Resources:** Sven B. Gould.

**Supervision:** Sven B. Gould.

**Visualization:** Parth K. Raval, Sven B. Gould.

**Writing – original draft:** Parth K. Raval, Alexander I. MacLeod, Sven B. Gould.

**Writing – review & editing:** Parth K. Raval, Alexander I. MacLeod, Sven B. Gould.

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
