## [Editor Report · Decision Letter 0]

2 Oct 2023

Dear Dr Gould, 

Thank you for submitting your manuscript entitled "A molecular atlas of plastid and mitochondrial adaptations across the evolution from chlorophyte algae to angiosperms" for consideration as a Research Article by PLOS Biology.

Your manuscript has now been evaluated by the PLOS Biology editorial staff, as well as by an academic editor with relevant expertise, and I'm writing to let you know that we would like to send your submission out for external peer review.

Once your full submission is complete, your paper will undergo a series of checks in preparation for peer review. After your manuscript has passed the checks it will be sent out for review. To provide the metadata for your submission, please Login to Editorial Manager (https://www.editorialmanager.com/pbiology) within two working days, i.e. by Oct 04 2023 11:59PM.

Kind regards,

Roli Roberts

Roland Roberts, PhD

Senior Editor

PLOS Biology

rroberts@plos.org

---

## [Decision Letter · Decision Letter 1]

22 Nov 2023

Dear Dr Gould,

Thank you for your patience while your manuscript "A molecular atlas of plastid and mitochondrial adaptations across the evolution from chlorophyte algae to angiosperms" was peer-reviewed at PLOS Biology. It has now been evaluated by the PLOS Biology editors, an Academic Editor with relevant expertise, and by three independent reviewers. 

In light of the reviews, which you will find at the end of this email, we would like to invite you to revise the work to thoroughly address the reviewers' reports.

You'll see that reviewer #1 likes the overall idea, calling it ambitious, but thinks that several conclusions are over-reaching and will need additional analyses to support them properly. S/he poses a number of questions, some of which relate to the methodology and some to the data used, each of which may be introducing significant bias (reliance on just four proteomes, have any mitochondrial proteins been lost in Archaeplastida, unwarranted inference of targeting from homology alone, sensitivity of clustering to artefact, need for more transcriptomes to overcome sampling bias); s/he has some constructive suggestions as to how to address these. Reviewer #2 is broadly positive, but has several queries about the language used, and wants you to tone down several claims and be more circumspect. Reviewer #3 enjoyed reading the paper, but has a number of concerns, including lack of clarity about the methods, the need for more proteomic datasets to identify MOGs/POGs (reviewer #1’s first point relates to this), and excessive “adaptationist” language.

Given the extent of revision needed, we cannot make a decision about publication until we have seen the revised manuscript and your response to the reviewers' comments. Your revised manuscript is likely to be sent for further evaluation by all or a subset of the reviewers.

**IMPORTANT - SUBMITTING YOUR REVISION**

*Re-submission Checklist*

*Published Peer Review*

*PLOS Data Policy*

*Blot and Gel Data Policy*

Sincerely,

Roli Roberts

Roland Roberts, PhD

Senior Editor

PLOS Biology

rroberts@plos.org

REVIEWERS' COMMENTS:

Reviewer #1:

This is an interesting, and ambitious attempt to reconstruct the organelle proteome innovations associated with the terrestrialisation of land plants. The authors show in particular the large number of key chloroplast and mitochondrial proteins associated with modern plants that originate within the Viridiplantae common ancestor, and also the zygnematophyte/ embryophyte common ancestor. The homology analyses are not particularly elaborate but are performed consistently across a large dataset of proteins and species, and can provide us with a lot of information about the stepwise accumulation of « plant-associated » organelle proteins.

My concerns are not so much to do with the methodology of the paper, or the results obtained, but unfortunately the conceptualisation and questions posed. These do not invalidate the results already obtained but would require substantial additional analysis to address well, or at the least a significant toning down of the conclusions within the paper. In brief :

- Organelle-associated proteins are defined by occurrence in at least one of four experimentally validated proteomes, three from plants and one from a green alga. This is reasonable (in the absence of validated plastid proteomes from red algae or glaucophytes) but naturally biases our understanding to what makes up a chloroplast to what is found in plants. Thus, the authors should be very cautious to suggest a gain in organelle proteome complexity in the land plant lineage : perhaps they just detect a greater proportion of the plastid proteomes due to closer homology to the experimental datasets used.

- An equally valid question for which good experimental data do exist : what mitochondrial proteins identified in non-photosynthetic eukaryotes are absent from archaeplastida ? Can the authors infer any trends concerning the loss of mitochondrial functions as a result of the acquisition of photosynthesis ? Provided the non-green species in the dataset include ones with experimentally resolved mitoproteomes, this information should already exist in the authors' data.

- Similarly, protein localisation can change over evolutionary timeframes and with evolutionary distance : dual-targeted proteins and protein families independently recruited to plastids and mitochondria (e.g., PPRs) are great examples. In the absence of either rigorous in silico or experimental localisations, the authors should not imply that evolutionary homology to one known plastid protein equates to plastid-associated functions across an entire protein orthogroup. This could be probed by custom in silico targeting predictors (e.g., PredAlgo), and/ or identification of coexpression trends in species for which extensive transcriptome data exist (i.e., genes coexpressed with others of known photosynthetic function might be expected to code for plastidial proteins)

- The clustering approach, while it sets thresholds reasonable for the detection of proteins shared across multiple green species, does not apply robust criteria for conservation of proteins in individual sub-lineages, and I am concerned that contamination or HGT into individual libraries considered might lead to an overestimation of the origin points. Beyond the subsampling approaches used, have the authors tried playing with the species thresholds (e.g., two green algae minimum required to assign Viridiplantae origin) to identify origin nodes ?

- In the opposite direction, while the authors have prioritised high-quality reference genomes, the paucity of sampling for specific groups (i.e., one lycophyte, glaucophyte, and palmophyllophyte species) are likely to underestimate the organelle protein complexity associated with these lineages, due to library incompletion and species-specific secondary loss. Particular concern are the glaucophytes given their relevance to assigning protein innovations found in the Viridiplantae common ancestor and secondary losses in Rhodophyta : here I feel the authors should judiciously augment their sampling with the more complete reference transcriptomes available for some species (e.g., Gloeochaete wittrockiana 1Kp and MMETSP transcriptomes, Cyanoptyche gloeocystis and Glaucocystis nostochinearum 1Kp transcriptomes)

Minor points :

- Line 34 : « chlorophytes, streptophytes and the prasinodermophytes that are thought to be the sister lineage to the two former » - is this universally agreed ? It is contradicted by the topologies shown in figs. S2 and S3

- Line 35 : it would be fair to note that some terrestrial chlorophytes (e.g., lichen symbionts) exist and are ecologically relevant

- Line 38 : similarly, polyplastidy is not exclusively the domain of terrestrial embryophytes, although it is frequently observed in multicellular and sessile groups (red algae, giant kelps)

- Line 53 : as a discrete (targeting) question per the above, it would be interesting to probe the embryophyte-associated orthogroups for peroxisomal targeting motifs, which may be comparatively easier to spot than cTPs or mTPs. Similarly, what proportion of the embryophyte-associated orthogroups have signal peptide positive predictions ? Evolution of the plastid has big impacts on the endomembrane system (dual-targeted proteins, rerouting of plastid glycoproteins through the Golgi, etc)

- Line 115 : by curiosity, have the authors tried to plot deviations in the ratio of POGs to MOGs recovered in individual species, to identify lineages with possible secondary reductions in plastid function ? I do not see any secondarily non-photosynthetic species in the authors' dataset but the inclusion of facultative mixotrophs (Auxenochlorella protothecoides, Chlamydomonas reinhardtii) could be used to probe if the loss of mixotrophy causes a simplification or elaboration of plastid metabolism.

- Line 121 : the mitochondrial biogenesis proteins identified as POGs would be interesting to probe at the level of protein targeting. Can the authors say anything about transit peptide length or charge, consistent with dual or ambiguous targeting, in any of these orthogroups ?

- Lines 168-172 : PPR proteins are certainly important for RNA processing in plants, but may have more limited roles in other lineages (I only know, for example, of one experimentally validated PPR in apicomplexans). Did the authors look at the distribution of OPRs, TPRs, or other adaptor proteins known to function in RNA processing in other algal groups ? What about more broadly the distribution and diversification of organelle RNA nucleases, which are of equal importance for regulating non-coding transcript accumulation in plant plastids and mitochondria ? 

- Line 184 : can the authors say anything about the predicted targeting sequences associated with the mTERF OGs ?

- RNA processing section : it would be good to discuss nucleus-encoded RNA polymerases as a known « control » for a green-specific plastid innovation. 

- Biochemistry section : one of the most classical differences between green and non-green plastids is the use of a plastid-encoded form ID rubisco in rhodophytes and glaucophytes. I would presume that the nucleus-encoded small subunit IB rubisco would be detected as a « Viridiplantae-associated gain » in the authors data, which while erroneous would be a good control for the robustness of the orthogroup reconstruction pipeline. I wonder also if there are any plastid-encoded functions in the red lineage that are either detected as Viridiplantae-associated gains (i.e., green-specific EGTs) or indeed are universally absent from Viridiplantae nuclear and organelle genomes.

- Plastid biogenesis section : where does peptidoglycan biosynthesis fall in the OG data ? This would be a plastid OG that should be shared between glaucophytes, KCC streptophytes, and bryophytes to the exclusion of embryophytes and (presumably) chlorophyte lineages

- Lines 279-285 : do any of these TOC proteins show tissue-specific paralogues in embryophytes ? Can the authors differentiate between possible « root » and « leaf »-associated isoforms ?

Reviewer #2:

The authors present a large analysis of gene content in diverse organisms, emphasizing plastid-associated genes. There seems to be supplementary data that I did not have access to, so I do not have an exhaustive list of the taxa that were analyzed, but I can estimate the taxonomic scope from the phylogenetic trees presented. Overall, the analysis seems to have been carefully conducted. 

The writing itself is sometimes awkward or grammatically incorrect; there are a number of sentence fragments, malformed clauses, peculiar word choices, and ambiguous turns of phrase. Sometimes this is merely distracting, but at times the intended meaning is genuinely difficult to infer. It appears that the first author is a junior scientist and is still developing their writing. That said, there are a few things that need to be cleaned up. Latin binomials (e.g. genus and species) need to be italicized, but family and above are not italicized. In many cases the binomials are not italicized. I personally found some of the colonial language jarring. Literally "aboriginal" means "from the beginning" but in context, I found the implication could be read as primitive which is possibly harmful. "Conquest of the land" implies both a volition in evolution and has an inherent violence. 

Importantly, the manuscript also includes language that is over-interpreting what is justified by the data. Wording is often credulous when there is substantial uncertainty. At times this extends to statements that are not supported by the underlying facts. For example, regarding mono- vs poly-plastidy, the description in the manuscript does not reflect the taxonomic distribution of polyplastidy from De Vries & Gould 2018. For example, when describing polyplastidy authors ignore the polyplastidy found in the stoneworts (Chara, Nitella, etc), as wel

---

## [Decision Letter · Decision Letter 2]

22 Mar 2024

Dear Dr Gould,

Thank you for your patience while we considered your revised manuscript "A molecular atlas of plastid and mitochondrial evolution from algae to angiosperms" for publication as a Research Article at PLOS Biology. This revised version of your manuscript has been evaluated by the PLOS Biology editors, the Academic Editor and the original reviewers.

Based on the reviews, we are likely to accept this manuscript for publication, provided you satisfactorily address the remaining points raised by the reviewers and the following data and other policy-related requests.

IMPORTANT - Please attend to the following:

a) Please change your Title to "A molecular atlas of plastid and mitochondrial proteins reveals organellar remodeling during plant evolutionary transitions from algae to angiosperms"

b) Please address the remaining requests from reviewers #1 and #3.

c) Please address my Data Policy requests below; specifically, we need you to supply the numerical values underlying Figs 1ABCDE, 2ABCD, 3ABCD, 4AB, 5, S2, S3ABCD, S4, S5, S6AB, S7AB, S8, S9 (many of these will be treefiles, I guess), either as a supplementary data file or as a permanent DOI’d deposition. I note that you currently provide raw data files as part of your bioRxiv deposition (https://www.biorxiv.org/content/10.1101/2023.09.01.555919v2.supplementary-material). This is not sufficient; please could you instead deposit this in a separate permanent DOI’d repository (e.g. in Zenodo) and provide this URL (see below).

d) Please cite the location of the data clearly in all relevant main and supplementary Figure legends, e.g. “The data underlying this Figure can be found in S1 Data” or “The data underlying this Figure can be found in https://zenodo.org/records/XXXXXXXX"

e) Please make any custom code available, either as a supplementary file or as part of your data deposition.

We expect to receive your revised manuscript within two weeks. 

*Published Peer Review History*

*Press*

Sincerely,

Roli

Roland Roberts, PhD

Senior Editor

rroberts@plos.org

PLOS Biology

DATA POLICY:

Regardless of the method selected, please ensure that you provide the individual numerical values that underlie the summary data displayed in the following figure panels as they are essential for readers to assess your analysis and to reproduce it: Figs 1ABCDE, 2ABCD, 3ABCD, 4AB, 5, S2, S3ABCD, S4, S5, S6AB, S7AB, S8, S9 (many of these will be treefiles, I guess). NOTE: the numerical data provided should include all replicates AND the way in which the plotted mean and errors were derived (it should not present only the mean/average values).

CODE POLICY

Per journal policy, as the code that you have generated is important to support the conclusions of your manuscript, we require that you make it available without restrictions upon publication. Please ensure that the code is sufficiently well documented and reusable, and that your Data Statement in the Editorial Manager submission system accurately describes where your code can be found.

DATA NOT SHOWN?

REVIEWERS' COMMENTS:

Reviewer #1:

I am happy with the revisions made by the authors to their manuscript, and their detailed and thoughtful responses to my questions. I have two additional points that I think can be improved in the paper:

1) I still stand by the point that the absence of experimentally validated organelle proteomes for red algae, alongside the lack of reliable targeting predictors for this group, limits our understanding of their composition and complexity. I would like the authors to explicitly discuss this, its implications for their data (e.g., POGs or MOGs unique to red algae, and not found in Chloroplastida) in their Discussion, and it would be helpful to suggest some possible experimental solutions (e.g., LOPIT on model red algal species, or proximity proteomics of transformable species such as Porphyridium with known plastidial markers).

2) The complexity of the secondary plastid proteomes in the authors' dataset is likely hugely underestimated, reflecting the evolutionary distance between these and Chloroplastida, and (for targeting predictions) the different pre sequences (e.g. with hydrophobic N-terminal targeting sequences) used to direct proteins to the plastids of these groups. This underestimation is in effect visible in Fig. 2 where the secondary plastid containing groups lack many of the 1y chloroplast-associated POGs, but show equivalent retrieval of MOGs as Archaeplastida. While not the focus of this study, I think the authors could address this readily by computing the overlap between their orthogroups and experimentally validated secondary plastid proteomes, e.g. Euglena (Novak Vanclova et al, 2020) or Phaeodactylum (Huang et al, 2024) to see how many of these secondary plastid-associated proteins escape incorporation into POGs assembled from a Chloroplastida-anchored dataset.

Reviewer #2:

The authors seem to have missed the fact that I liked the paper. The changes have conscientiously addressed my primary concerns. Where we disagree is well within the range of authorial discretion. I am now comfortable recommending the paper for publication.

Reviewer #3:

The authors have addressed my concerns and I am happy to recommend this revised version.

I still have a (very) minor comment about the use of the word "adaptation"

Line 25 In the abstract, I suggest removing the word 'adaptation' (already used line 17) in "The analyses of gained orthogroups identifies molecular adaptations", as a demonstration that these gains are adaptations may be controversial (while I agree with some examples detailed later in the manuscript). A more correct phrasing of this sentence would be "The analyses of gained orthogroups identifies molecular changes of organelle biology that connect to the diversification of major lineages and facilitated major transitions from chlorophytes en route to the global greening and origin of angiosperms."

---

## [Editor Report · Decision Letter 3]

28 Mar 2024

Dear Dr Gould,

Thank you for the submission of your revised Research Article "A molecular atlas of plastid and mitochondrial proteins reveals organellar remodeling during plant evolutionary transitions from algae to angiosperms" for publication in PLOS Biology. On behalf of my colleagues and the Academic Editor, Andrew Tanentzap, I'm pleased to say that we can in principle accept your manuscript for publication, provided you address any remaining formatting and reporting issues. These will be detailed in an email you should receive within 2-3 business days from our colleagues in the journal operations team; no action is required from you until then. Please note that we will not be able to formally accept your manuscript and schedule it for publication until you have completed any requested changes.

Sincerely, 

Roli Roberts

Senior Editor

PLOS Biology

rroberts@plos.org